# Study of the Effect of Modification of Asphalt on the Rheological Properties Employing Microwave Radiation—An Aging Study

Khalid Ahmed Owaid [1], Raghed Y. Ghazal [1] and M. A. Abdelzaher [2,*]

1   Department of Chemistry, Education College for Pure Science, Mosul University, Mosul 41002, Iraq
2   Environmental Science and Industrial Development Department, Faculty of Postgraduate Studies for Advanced Sciences, Beni-Suef University, Beni-Suef 62511, Egypt
*   Correspondence: m.abuelseoud@psas.bsu.edu.eg; Tel.: +2-010-6800-3341

**Abstract:** This study focuses on producing asphalt with improved rheological properties that differ from the original asphalt and are less affected by aging conditions. The rheological properties of Qayara asphalt were enhanced by modifying the asphalt using spent rubber tire (SRT) with different percentages of anhydrous aluminum chloride. Percentages ranging from 1.0% by weight of the spent tire rubber were added after proceeding with the thermal crushing process. The percentages of anhydrous aluminum chloride catalyst were 0.4 and 0.8%, respectively. This mixture was microwaved at 270 watt of power for 4, 8, and 12 min, respectively. The measurements performed are plasticity, penetration, softening point, and penetration index. The previously mentioned measurements were also made on the modified asphalt one year after the modification process to understand the effect of aging conditions. The microstructure and thermodynamics have been characterized by FE-SEM and EDX measurements. This study provides good rheological properties of the modified bitumen binder that is aging-resistant.

**Keywords:** modified bitumen binder; spent rubber; tire microwave oven and aging resistant





## 1. Introduction

The asphalt industry is one of the most important industries that exist today. Asphalt is a thermoplastic material and keeps its condition very well. At normal temperatures, asphalt is a liquid or semi-solid material with a high viscosity and works as an excellent insulator from water that is not affected by any factors such as salts or even acids [1–4]. It can be black or dark brown in color. Asphalt is a heavy hydrocarbon component derived from the direct distillation of crude oil, and it is one of the best materials used in the process of paving roads in streets, airfields roads, and others [5]. It also consists of three main layers that serve as the basic layers, namely, the base layer, the bonding layer, and the third layer is the surface layer. The first layer consists of ordinary gravel or crushed gravel, as well as sand and filler. The second layer, consisting of crushed gravel, its proportion is about 90%, and the thickness of that layer is 5 cm up to 8 cm. The last layer, which is the surface layer, consists of the same components that make up the second layer but with different sizes and thicknesses, from 6 cm to 8 cm in thickness [6–10]. Obtaining asphalt with rheological properties and resistance to aging factors is an important and necessary issue to ensure the longest possible period for asphalt use in the most important area, which is paving [3,11]. Asphaltic materials are heterogeneous hydrocarbon materials produced by direct distillation that contain sulfur, nitrogen, and oxygen (S, N, O), in addition to cyclic and non-cyclic compounds [12]. One of the most important methods for obtaining asphalt with rheological properties that exceed those of the original asphalt is the rheological modification of asphalt processes using polymeric materials [13]. Chemical modification with polymers was chosen over other methods because the produced asphalt has compatible properties [14–17]. Due to the importance of asphalt and its use in the field

of paving and its availability in large quantities, many researchers have modified asphalt and improved its rheological properties by using various additives, such as Mashaan and Nikraz, who studied the engineering properties of asphalt binder modified with local waste of polyethylene terephthalate (PET), a common type used for local road surfaces in Australia [12,18]. Several tests were conducted through dynamic shearing, thin film furnace, and aging tests for modified asphalt, and it was found through the study that the best percentage of re-polymer is (6.0–8.0%) of the weight of the mixture, which can improve the properties of the concrete paving mixture [18,19].

Cunha et al. studied the rheological properties of modified asphalt containing natural fibers sourced from Amazon rain forests, discovering that these additives significantly improve physical properties, such as fluidity, using TG and DSR tests [20,21]. Saleh studied the effect of adding the different ratios of asphaltene to Begi and Qayara asphalt [22]. He measured the rheological properties, and the results obtained after 18 months of aging, compared with those of untreated asphalt, showed an increase in the homogeneity system for both Qayara and Begi asphalt, with improvement in the aging specifications for Qayara asphalt only [23–25]. Anjan and Veeragavan found that the mixing of asphalt with certain additives caused to improve the mechanical properties of the studied asphalt [26]. Hamedi and Joubani studied the effect of styrene-butadiene rubber-modified asphalt (SBR) on improving the moisture susceptibility of asphalt mixtures. Repeated loading tests were performed in wet and dry conditions side by side, and the study showed that the presence of this polymer in the mixture caused a positive change, as it increased the strength of the mixture against moisture damage [27]. Nikraz and Mashaan studied the engineering properties of the asphalt with local points (Ethylene Terephthalate) evaluated from tests, dynamic shear test, and aging test, and the study showed that the best percentage of re-polymer is (2.0 to 6.0%) of the best possible modification [28]. Pakenari and Hamidi were also able to study some of the rheological properties of a warm recycled asphalt mixture (WAM). The results showed that the increase in the time of placing the mixture in the kiln increased the hardness coefficient, which reduced the stress life and increased the adhesion between the asphalt binders and rubble [29,30]. Due to the urgent need to produce materials of great feasibility and economic value, the production of asphalt with specifications differing from the specifications of the basis of traditional asphalt materials and suitable for use in areas not commensurate with the normal use of conventional asphalt is necessary.

The main objective of the current research was to evaluate the rheological properties of modified asphalt binders. Recycling old rubber tires reduces the amount of new bitumen needed in the paving process, reducing reliance on or importing new bitumen. Recycling reduces waste tire rubber, as the recycled material is no longer sent to the landfill as a huge accumulated quantity. Accordingly, with regard to the issue of estimating the amount of waste rubber tires in the province of Iraq, we attempted to develop an approach based on the principle of tons and proportionality—a process between the aforementioned in terms of approximate tons, shown in Figure 1, as an indicator of the expected percentage of the amount of waste rubber tires generated, which amounts to the total waste percentage for one year (2020), during the time period [31]. Cities with high RTW consumption are represented in the blue column, while those with low consumption are represented in the green bars. The green bars are the cities that spent less than 20 tons of SRT, and the blue bars are the cities that have the most.

Recycling the spent rubber tires and the asphalt pavement recycling process helps conserve natural resources and be sustainable. The use of modified asphalt by polymers is increasing at the present time because this method very useful for the environment because it uses polymeric solid wastes, which are polluting the environment. The aim of this study is to change these harmful substances into useful ones. Therefore, asphalt without modification is not climate resistant and needs improvement. Accordingly, asphalt modified by such means has considerable resistance to the process of aging.

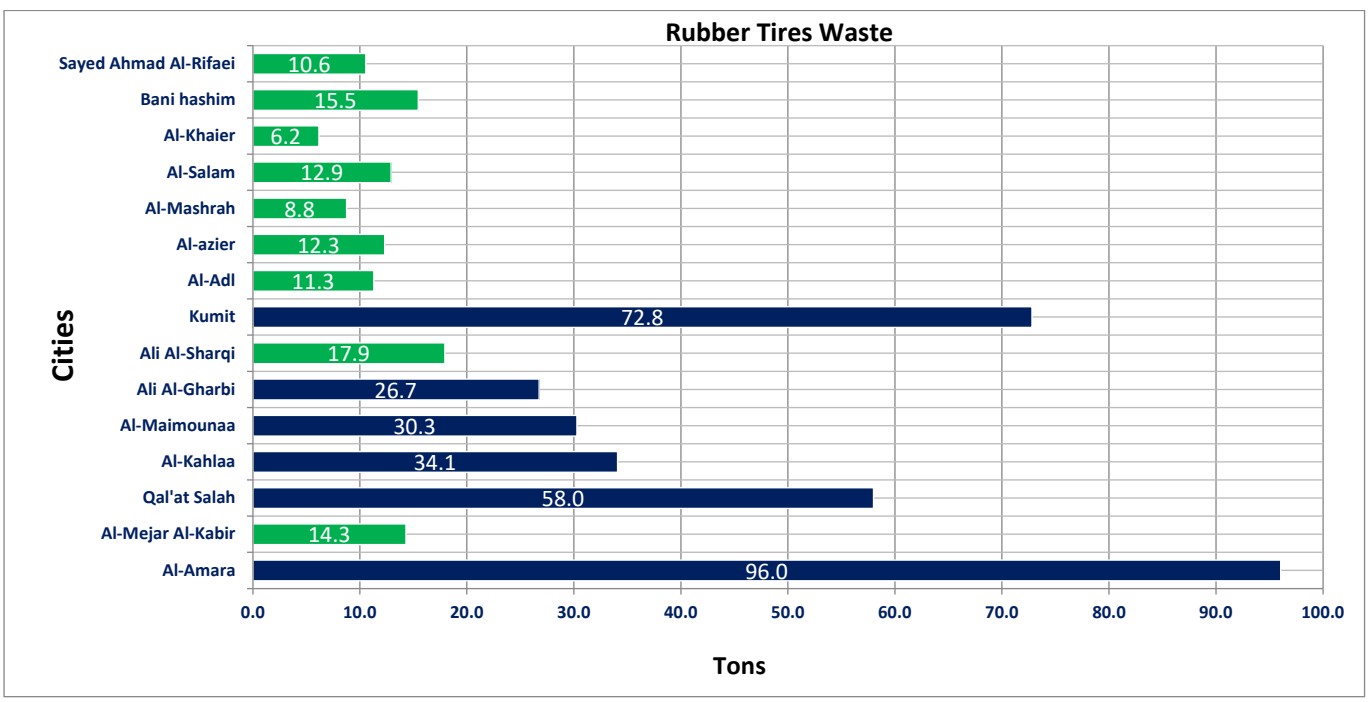

**Figure 1.** Accumulations of spent rubber tires for each Iraqi city.

## 2. Materials and Methods

### 2.1. Materials

The main raw materials were provided from Qayara crude asphalt, with the specs tabulated in Table 1. Bitumen binder was produced in Al-Qayara refinery Co., about 50 km south of Mosul city (North Baghdad, Iraq). Figure 2 shows the visual inspection of milled rubber tires (before and after), which were ground up to 1.0 μm mesh size (the content of isoprene rubber is 40%). The reclaimed rubber tires were provided by the General Company of the Babylon tire industry (Babylon, Iraq) [32]. Anhydrous aluminum chloride was supplied by Fluka Company (Darmstadt, Germany) for chemicals. Table 2 reports the specifications for Anhyd-AlCl$_3$. Twenty kilograms were collected from Qayara crude asphalt followed by homogeneity and dried at 105 °C for 24 h before practical work. Additionally, the Rheological properties of Qayara crude asphalt vs. the standard testing measurements (JTG E20, 2011) are detailed in Table 3 [33].

**Table 1.** Rheological properties of bitumen binder (Qayara).

| Rheological Properties | Minimum | Maximum | Mean |
|---|---|---|---|
| Softening point (°C) | 51 | 60 | 55 |
| Penetration (100 gm, 5 s, 25 °C) | 41 | 47 | 44 |
| Degree of Ductility (cm, 25 °C) | 10 | - | 10 |

**Table 2.** Technical specification of anhydrous aluminium chloride (as received).

| Specification | Anhydrous Aluminium Chloride (AlCl$_3$) |
|---|---|
| Molecular weight | 133.34 g/mol |
| Appearance (Color) | White |
| Features | Anhydrous, powder, 99.99% trace metals basis |
| Purity | 95%. |

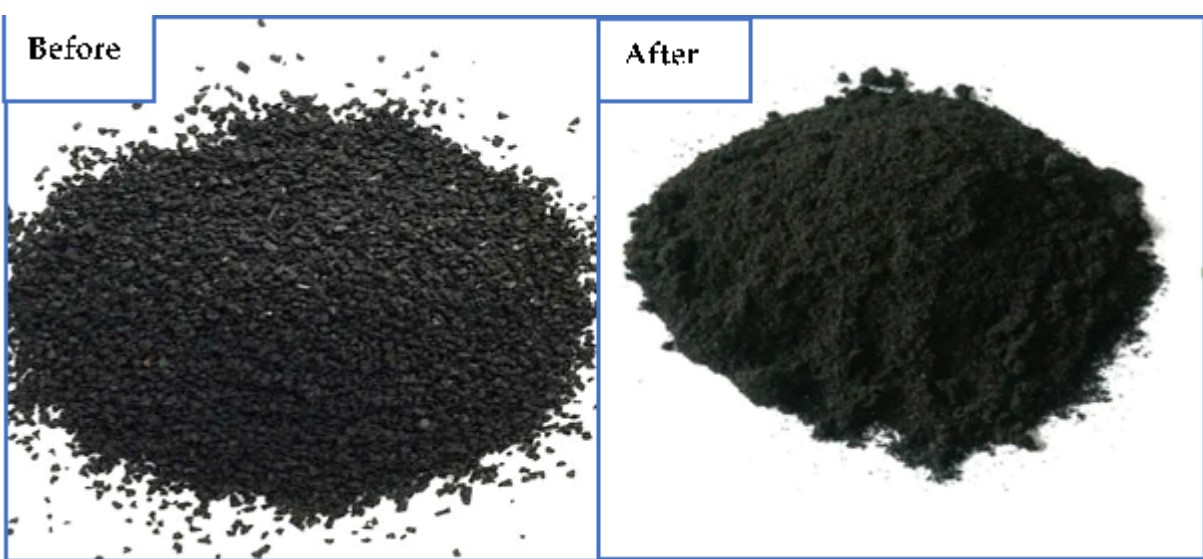

**Figure 2.** Visual inspection of milled rubber tires (before and after milling).

**Table 3.** Rheological properties of Qayara crude asphalt vs. the standard testing measurements (JTG E20,2011).

| Rheological Properties | Qayara Crude Asphalt | | The Standard Testing Measurements (JTG E20,2011) [33] | |
|---|---|---|---|---|
| | Minimum | Maximum | Minimum | Maximum |
| Softening point (%) | 51 | 60 | 55 | 65 |
| Penetration (100 gm. 5 s. 25 °C) | 41 | 47 | 20 | 40 |
| Degree of Ductility (cm. 25 °C) | 10 | - | 15 | - |

## 2.2. Materials Preparation and Methods

We developed six suggested mixtures for the current laboratory work. A total of 250 gm asphalt was thermally treated at 100 °C. Crushed reclaimed rubber tires at 1.0% wt. and catalyst ratios of both 0.5% and 1.0% were totally mixed with bitumen binder; mix proportions are shown in Table 4. The prepared samples were placed in an oven at 360 °C for 5, 10, and 15 min, respectively, for thermal homogeneity.

**Table 4.** Mix proportions of SRT incorporated with asphalt patches at different $AlCl_3$ ratios.

| Mix Proportions | | | | | | | | | | | |
|---|---|---|---|---|---|---|---|---|---|---|---|
| Mix Title | B0 | BR-1 | | BR-2 | | BR-3 | | BR-4 | | BR-5 | |
| Asphalt by weight (%) | 100 | 98.5 | 98.0 | 97.5 | 97.0 | 96.5 | 96.0 | 95.5 | 95.0 | 94.5 | 94.0 |
| Spent Rubber Tires (SRT) by weight (%) | - | 1.0 | 1.0 | 2.0 | 2.0 | 3.0 | 3.0 | 4.0 | 4.0 | 5.0 | 5.0 |
| Anhydrous Aluminum Chloride | - | 0.5 | 1.0 | 0.5 | 1.0 | 0.5 | 1.0 | 0.5 | 1.0 | 0.5 | 1.0 |
| Time, min. | 5.0 10.0 15.0 | 5.0 10.0 15.0 | 5.0 10.0 15.0 | 5.0 10.0 15.0 | 5.0 10.0 15.0 | 5.0 10.0 15.0 | 5.0 10.0 15.0 | 5.0 10.0 15.0 | 5.0 10.0 15.0 | 5.0 10.0 15.0 | 5.0 10.0 15.0 |
| Temp.°C | 360 °C | | | | | | | | | | |

In order to obtain a polymer with lower molecular weight, reclaimed rubber tires were exposed to mechanical and thermal crushing conditions before interaction with

bitumen binder material. The process was conducted based on the thermo-gravimetric analysis of the rubber tires by placing rubber tires in an oven at 360 °C, followed by cooling at ambient temperature and crushing by manual mortar. The specimen's rheological properties were measured, e.g., ductility test at 25 °C [34], penetration [35], softening point [36], and penetration index [37], as clearly identified elsewhere [38]. Two ratios were taken from reclaimed crushed rubber tires, with granules 1.0 mm mesh size, placed in a ceramic crucible, and covered with aluminum foil, then the crucible was thermally treated. As the temperature increased, the weight loss increased and vice versa; the weight loss increased from 39.03% (100 °C) to 99.07% (600 °C). The effect of aging under atmospheric conditions was carefully evaluated by measuring the penetration, ductility test at 25 °C, softening point, and penetration index; the cured specimen was kept for 365 days, and the measurement was repeated after a year. Figure 3 shows the specimen after curing for one year. In our study, a Marshal examination of the original model was conducted, in addition to one of the best samples before and after the aging procedure, which confirmed the specifications of the standard testing measurements (JTG E20, 2011) [33].

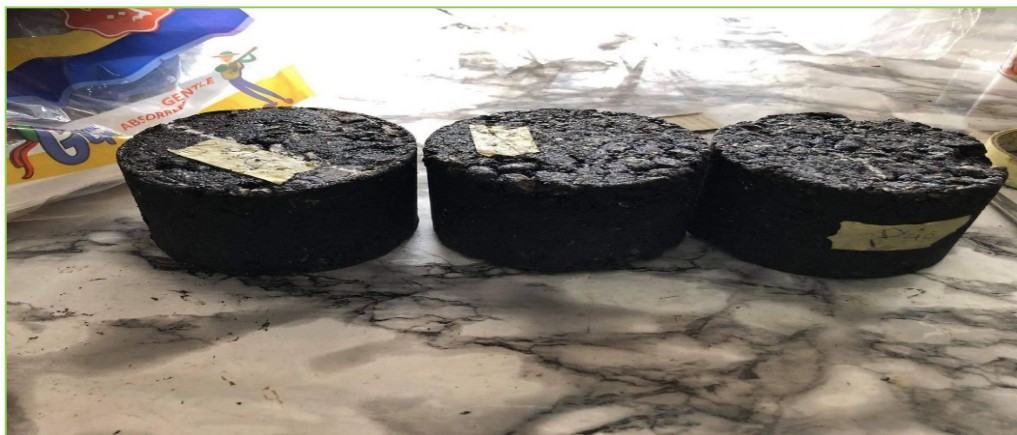

**Figure 3.** The specimen after curing for one year.

*2.3. Instrumentation*

A microwave oven (900 Watt, 2500 MHz- German, Tokiwa) was used to conduct the rheological modification of bitumen binder with SRT. The Ductility measurement was conducted according to the US standards of ASTM (D5-83) [39]. The measurement of the asphalt penetration was performed according to the ASTM (D36-70), using a penetrometer to comply with the norm explained elsewhere [35]. A ring and ball apparatus was used to measure the softening point in accordance with the specifications of ASTM (D5-85) universally adopted [40]. The morphology, e.g., FSEM of the specimen, was conducted with (FEI Company, The Netherlands) integrated with EDXA, namely, "an energy dissipation X-ray analyzer".

## 3. Results and Discussion

*3.1. Use of 0.5% Catalyst as Rheological Modifier of Asphalt*

SRT incorporated with asphalt in the presence of a catalyzed amount of anhydrous aluminum chloride by 0.5% ratio is shown in Figures 4–6. Three patches were tested for each examined mix, and the mean value was recorded. The rheological properties of the modified bitumen pastes catalyzed by 0.5% from anhydrous aluminum chloride at different times (5, 10, 15 min), respectively; 0.5% by weight of catalyst made the asphalt system more homogenous. The anhydrous aluminum chloride ($AlCl_3$) was selected as one of the Lewis acid catalysts for such interactions based on previous findings in the literature for the active role played by the catalyst in various industrial processes, such as the alkylation process and the stimulating effect in such operations [41].

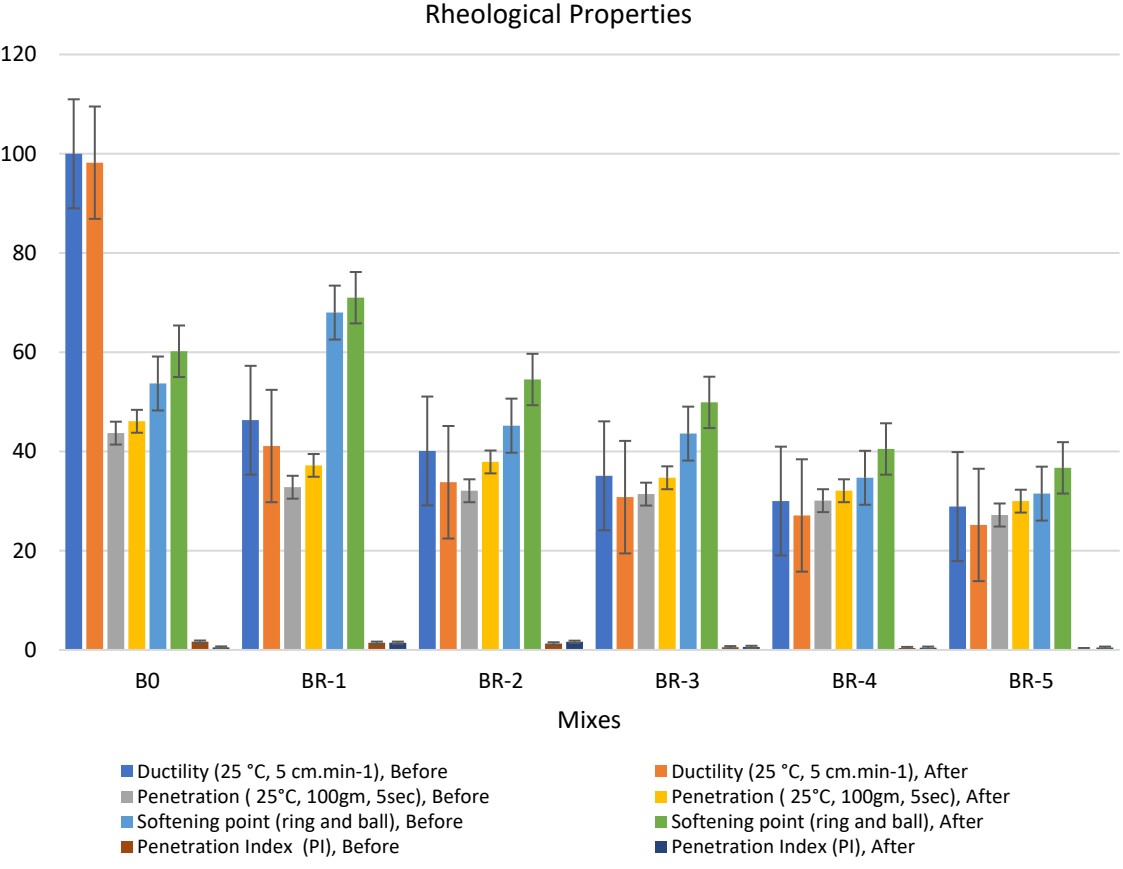

**Figure 4.** Properties of asphalt modified by 0.5% by weight of AlCl$_3$, using 1.0% of SRT before and after the aging procedure (at 5 min), treated at 360 °C.

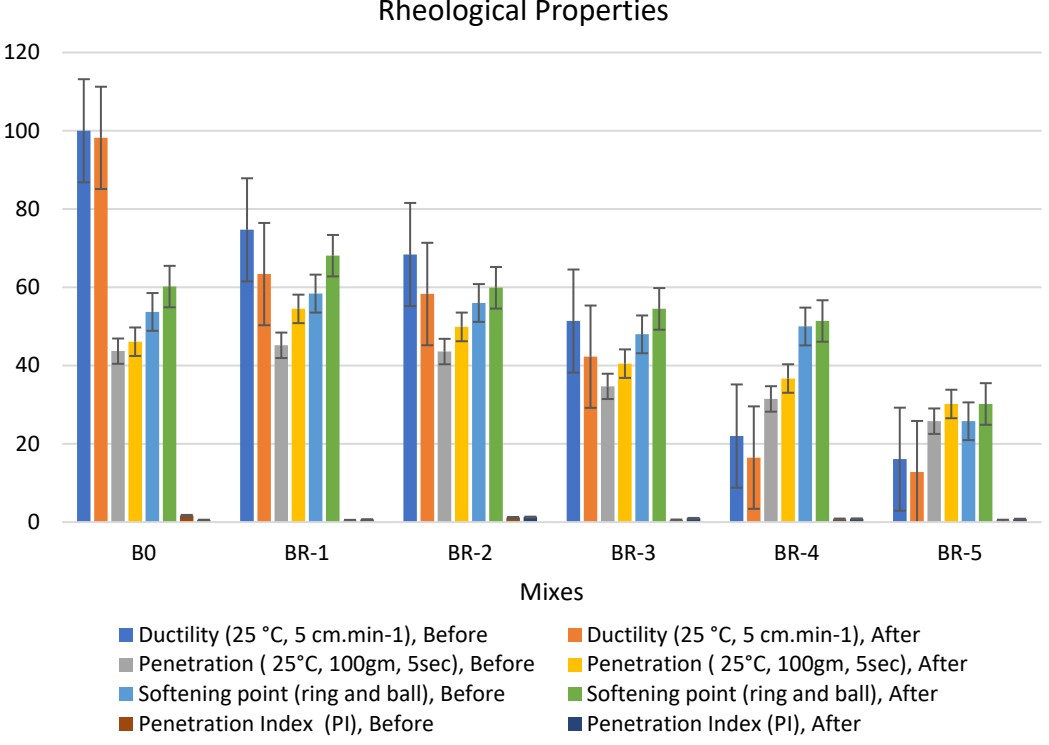

**Figure 5.** Properties of asphalt modified by 0.5% by weight of AlCl$_3$, using 1.0% of SRT before and after the aging procedure (at 10 min), treated at 360 °C.

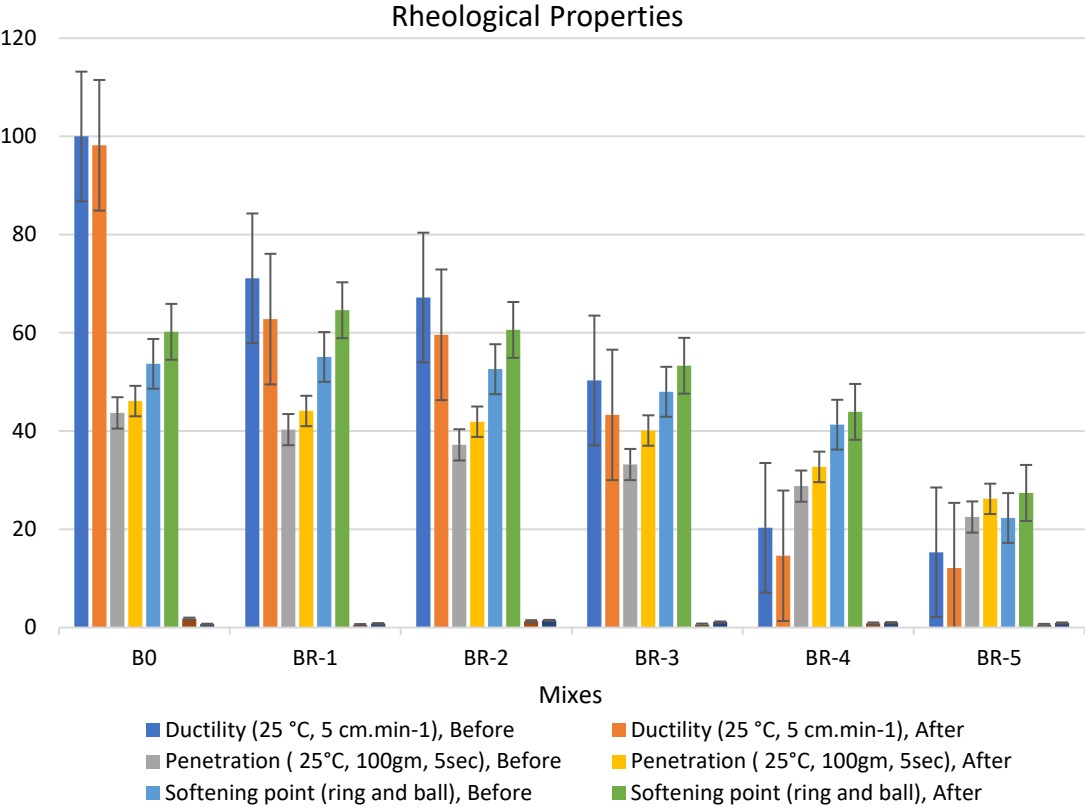

**Figure 6.** Properties of asphalt modified by 0.5% by weight of AlCl$_3$, using 1.0% of SRT before and after the aging procedure (at 15 min), treated at 360 °C.

Adding 0.5% of the AlCl$_3$ catalyst to the bitumen binder modified its physical properties due to the filling effect of SRT fine powder and increased the viscosity of the samples. The rheological properties of the specimens had the following order: BR1 > B0 > BR2 > BR3 > BR4 > BR5. The modified samples showed an increase in softening point values after aging (10 min), e.g., BR1 (68.1) > B0 (60.8) > BR2 (59.9) > BR3 (54.5) > BR4 (51.4) > BR5 (30.2). In addition, low values of penetration were detected after aging (10 s), BR1 (54.5) > B0 (46.1) < BR2 (49.6) > BR3 (40.5) > BR4 (36.7) > BR5 (35.2). Ductility decreased due to the filling effect. The penetration index was reduced because of the low penetration of BR1 compared to B0 due to increased bulk density and decreasing total porosity. Ductility (25 °C, 5 cm·min$^{-1}$) after aging (10 min) sharply decreased with an increase in SRT powder and decreasing the bitumen content, e.g., BR1 (63.4) < B0 (98.2) > BR2 (58.3) > BR3 (42.3) > BR4 (16.5) > BR5 (12.8). Rheological properties after aging at 5 min and 15 min reported low workability and operating performance because 5 min was not sufficient time to modify the properties, and at 15 min, most physical bonds of N-S-C were destroyed, weakening the specimen flexure pressure.

### 3.2. Use of 1.0% Catalyst as Rheological Modifier of Asphalt

SRT incorporated with asphalt in the presence of a catalyzed amount of anhydrous aluminum chloride in a 1.0% ratio is shown in Figures 7–9. The rheological properties of the modified bitumen catalyzed by 1.0% from anhydrous aluminum chloride at different times (5, 10, 15 min), respectively.

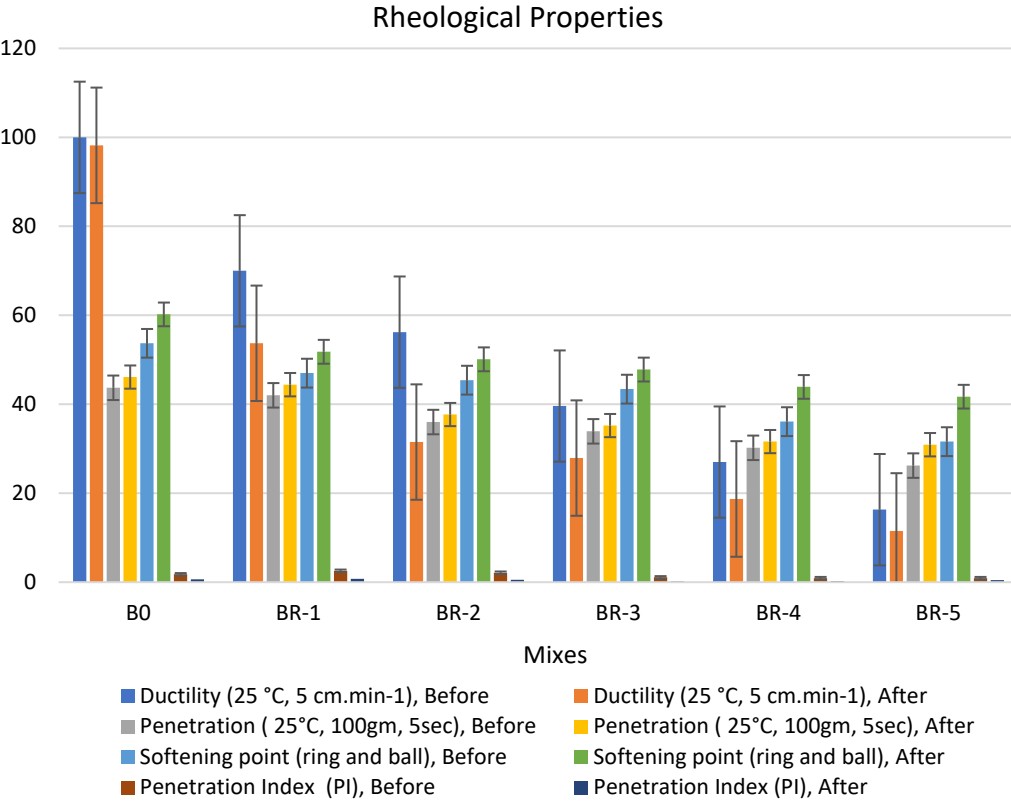

**Figure 7.** Properties of asphalt modified by 1.0% by weight of AlCl₃, using 1.0% of SRT before and after the aging procedure (at 5 min), treated at 360 °C.

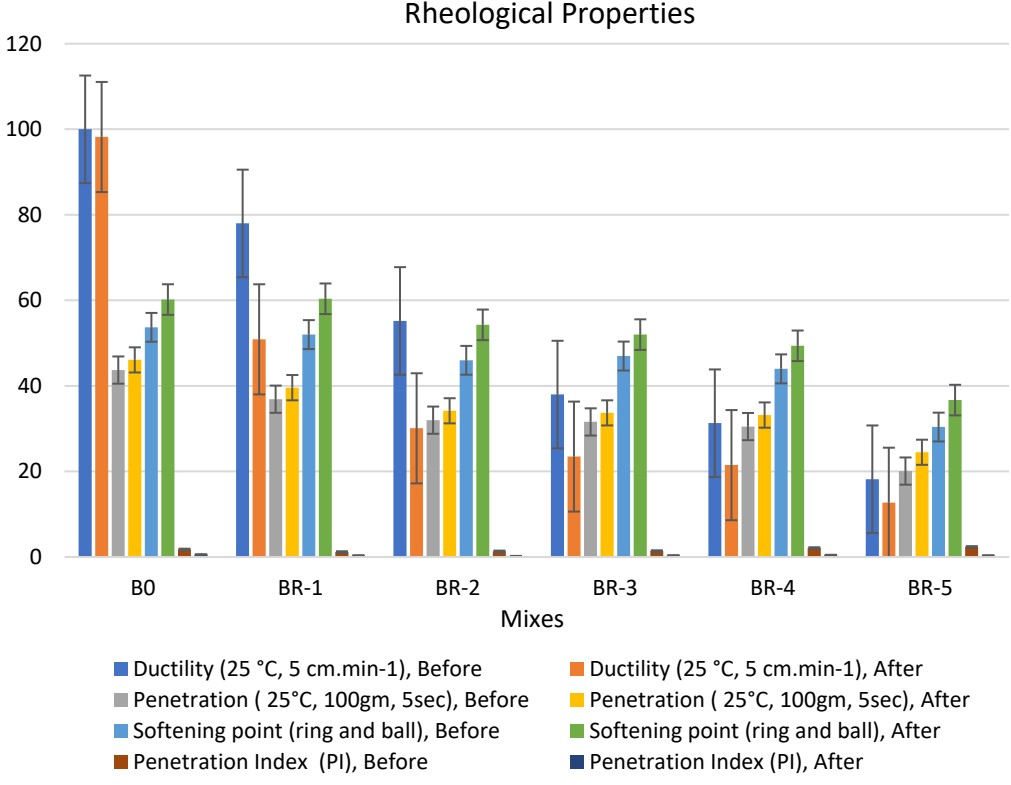

**Figure 8.** Properties of asphalt modified by 1.0 by weight of AlCl₃, using 1.0% of SRT before and after the aging procedure (at 10 min), treated at 360 °C.

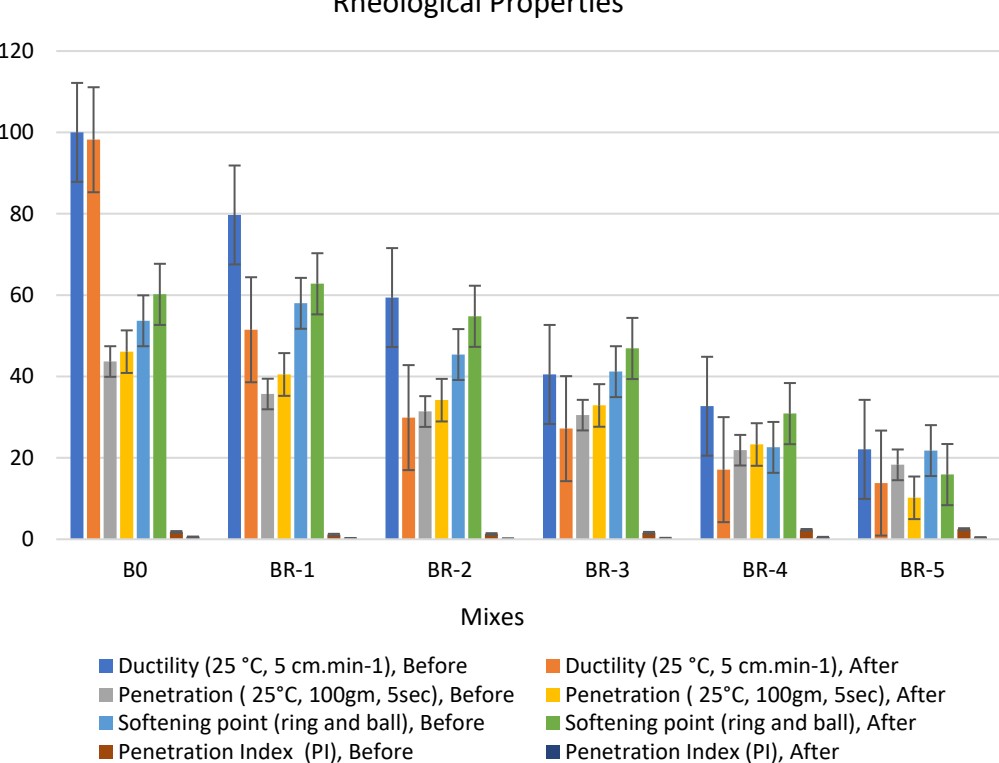

**Figure 9.** Properties of asphalt modified by 1.0% by weight of AlCl$_3$, using 1.0% of SRT before and after the aging procedure (at 15 min), treated at 360 °C.

Furthermore, the AlCl$_3$ catalyst was increased to 1.0% of the bitumen binder to enhance the physical properties. The filling effect of SRT fine powder increased the viscosity of the proposed samples. The rheological properties of the specimens have the same following order: BR1 > B0 > BR2 > BR3 > BR4 > BR5. The modified samples show an increase in softening point values after aging (10 s), e.g., BR1 (60.4) > B0 (60.2) > BR2 (54.3) > BR3 (52.0) > BR4 (49.4) > BR5 (36.7). In addition, low values of penetration were detected after aging (10 min), BR1 (39.6) < B0 (46.1) < BR2 (34.2) > BR3 (33.7) > BR4 (33.2) > BR5 (24.5). Ductility decreased due to the filling effect. The penetration index was reduced because of the low penetration of BR1 compared to B0 due to increased paste bulk density and decreasing total porosity. Ductility (25 °C, 5 cm·min$^{-1}$), after aging (10 s), sharply decreased with an increase in SRT powder and decreasing the bitumen content, e.g., BR1 (50.9) < B0 (98.2) > BR2 (30.1) > BR3 (23.5) > BR4 (21.5) > BR5 (12.7). Rheological properties after aging at 5 min and 15 min report low workability and operating performance because 5 min is not sufficient time to modify the properties, and at 15 min, most physical bonds of N-S-C are destroyed, leading to weak specimen flexure pressure.

### 3.3. Marshall Test

A group of three samples was tested for each composite. Figures 10 and 11 report the Marshall test results for selected specimens show better rheological properties. Three patches were tested for each examined mix, and the mean values were recorded. The examination indicated the suitability of the bitumen binder for paving by applying pressure (mechanical stress) on the specimen to be tested, and when the specimen began to deform, the stability and flow measurements were taken through certain gradients present in the device [42–45]. We noted that mixtures of asphalt modified by 0.5% and 1.0% by weight of catalyst AlCl$_3$, using 1.0% of SRT after the aging procedure (at 10 min), treated at 360 °C, reported good rheological properties.

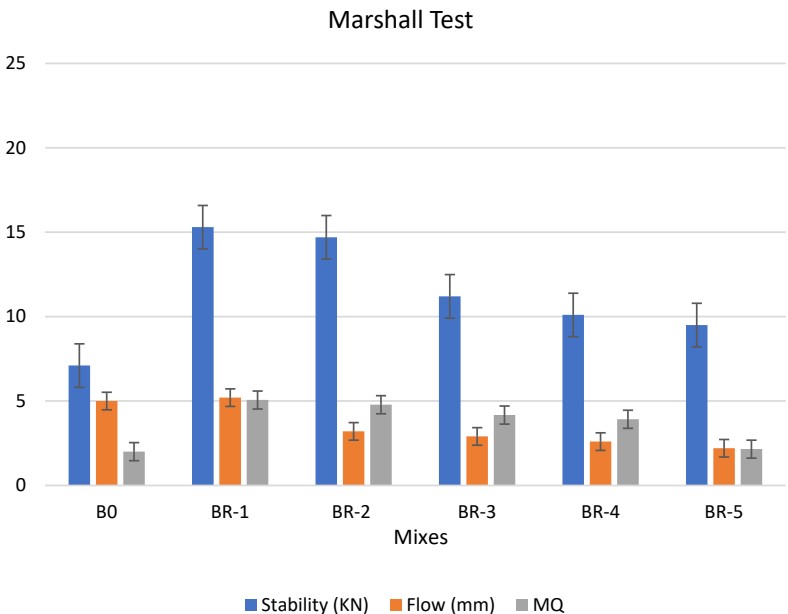

**Figure 10.** Marshall test of asphalt modified by 0.5% by weight of AlCl$_3$, using 1.0% of SRT after the aging procedure (at 10 min), treated at 360 °C.

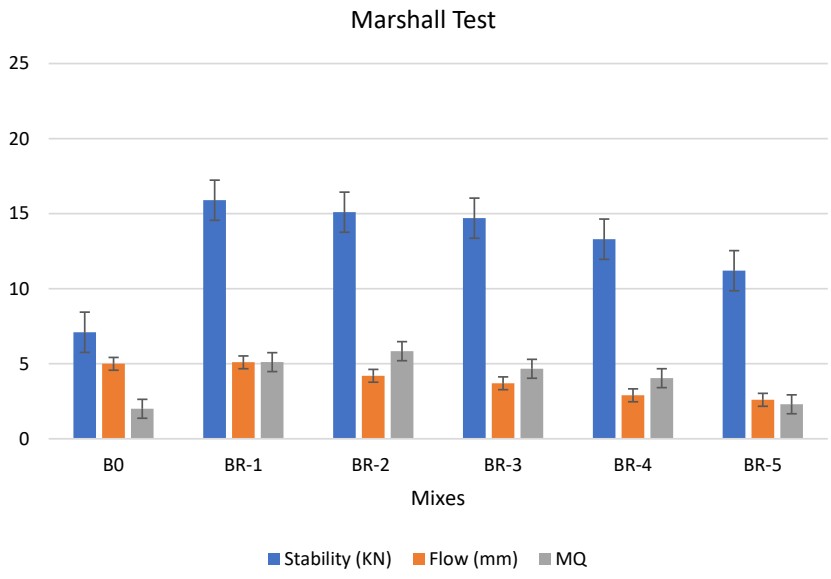

**Figure 11.** Marshall test of asphalt modified by 1.0% by weight of AlCl$_3$, using 1.0% of SRT after the aging procedure (at 10 min), treated at 360 °C.

It is clear from the above figure that the modified specimen is better than the original asphalt (#B0) in terms of stability and creep values if they could be used in asphalt paving. Additionally, we also noted that the specimen was more stable and unaffected by the aging conditions through the values of stability and creep, which were measured according to the Marshall test. This is a good indication of the pavement's ability to resist deformation and distortion caused by climate change and repeated transportation loads on the roads.

We concluded that the modified specimen by 0.5% and 1.0% with weight of AlCl$_3$, using 1.0% of SRT after the aging procedure (at 10 min), treated at 360 °C, are better than the original asphalt (#B0), in terms of stability and flow values if they could be used in asphalt paving. The Marshall test of the specimens had the same following order: BR1 > B0 > BR2 > BR3 > BR4 > BR5. 0.5% by weight of AlCl$_3$-catalyst after the aging procedure (at 10 min) shows higher performance than 1.0% by weight of AlCl$_3$ at the

same aging conditions; this may be attributed to 1.0% catalyst is sufficient for enhancing the physical properties of the mixture after 365 days of curing [44]. The specimen was more stable and unaffected by the aging conditions at 0.5% catalyst, through the values of stability: BR1 (15.3) > B0 (7.1) > BR2 (14.7) > BR3 (11.2) > BR4 (10.1) > BR5 (9.5). Regarding the followability, which was measured according to the Marshall test, shows the following order BR1 (5.2) > B0 (5.0) > BR2 (3.2) > BR3 (2.9) > BR4 (2.6) > BR5 (2.2). This is a good indication of the pavement's creep and ability to resist deformation and distortion caused by climate change and repeated transportation loads on the roads.

### 3.4. The Conventional Performance

Eventually, as tabulated in Table 5, the degree of ductility, penetration, softening point, and extension values for the asphalt modified by 0.5% and 1.0% by weight of $AlCl_3$, respectively, using 1.0% of SRT after the aging procedure at different periods, treated at 360 °C. Composites recorded results within the limits of specification for bitumen shown in Section 2.1. The ductility values decrease at 0.5% and 1.0% by weight of $AlCl_3$, respectively, by weight of the SRT additive. Nevertheless, it remained within acceptable limits [46,47]. As the replacement ratio of SRT increases, it reduces the conventional performance of asphalt with aging. Friction should be as little as possible with the asphalt to improve age ability. High bulk density and low porosity process increase the extension effect, but with the SRT addition effect, reduce the extension effect and penetration results. It is reported that the penetration results of the modified asphalt composites, in the order AS > BR-1 < BR-2, have better results because the extra addition of SRT from either source or bio-fuel shows a negative performance when added to modified asphalt.

**Table 5.** Conventional properties of asphalt modified by 0.5% and 1.0% by weight of $AlCl_3$, using 1.0% of SRT after the aging procedure (at 10 min), treated at 360 °C.

| Mix Title | Conventional Properties | | | | | | | | | |
|---|---|---|---|---|---|---|---|---|---|---|
| | Ductility (cm) | | Softening Point | | Penetration | | Penetration Index | | Asphaltens % | |
| $AlCl_3$ | 0.5% | 1.0% | 0.5% | 1.0% | 0.5% | 1.0% | 0.5% | 1.0% | 0.5% | 1.0% |
| AS | >150 | | 51 | 51 | 42.3 | 42.3 | −1.461 | −1.461 | 20.1 | 20.1 |
| BR-1 | >150 | | 53 | 53 | 41.5 | 41.9 | −0.670 | −0.888 | 24.1 | 24.5 |
| BR-2 | >150 | | 56 | 58 | 42.2 | 43.2 | −0.655 | −0.915 | 26.6 | 26.9 |
| BR-3 | >150 | | 56 | 59 | 42.9 | 44.1 | −0.631 | +0.223 | 28.1 | 30.2 |
| BR-4 | >150 | | 58 | 59 | 43.8 | 44.8 | −0.596 | +0.447 | 28.5 | 32.5 |
| BR-5 | >150 | | 59 | 63 | 44.7 | 46.2 | −0.337 | +0.673 | 29.2 | 33.2 |

### 3.5. Field Emission Scanning Electron Microscopy

Figure 12 FE-SEM of the asphalt sample as received from Qayara Refinery Company. The grade of asphalt depends on the raw source and chemical compositions, especially the carbon (C) and hydrogen (H) content. The physical properties of the asphalt were determined by the chemical composition of green circle hydrocarbon, which was substantiated by EDX analysis [48–50]. The green circle shows the sulfur content, as proven by EDX analysis. The content of sulfur shows the high purity of crude used and less contaminant present in the delivered mixes.

Field Emission Scanning Electron Microscopy (FE-SEM) was additives. Figure 13 report the microstructure for BR1 samples modified by 0.5% and 1.0% conducted on some selected samples obtained from modulating asphalt with polymeric catalyst after aging (10 min). The incorporation of polymer into the microstructure of the asphalt system further complicates the study of the internal structure of the modified asphalt and its effect on the rheological and morphological properties of asphalt because the shape and rheological properties of asphalt result from the interactions of polymer and asphalt [51,52]. In $BR1_{0.5}$ morphology is a more compact surface and has less porosity; there is a difference in the structural form of the conventional asphalt through SEM and EDX examination. $BRT_{1.0}$

morphologies have a distortion in the internal skeleton structure, which decreases the compression pressure and MQ results.

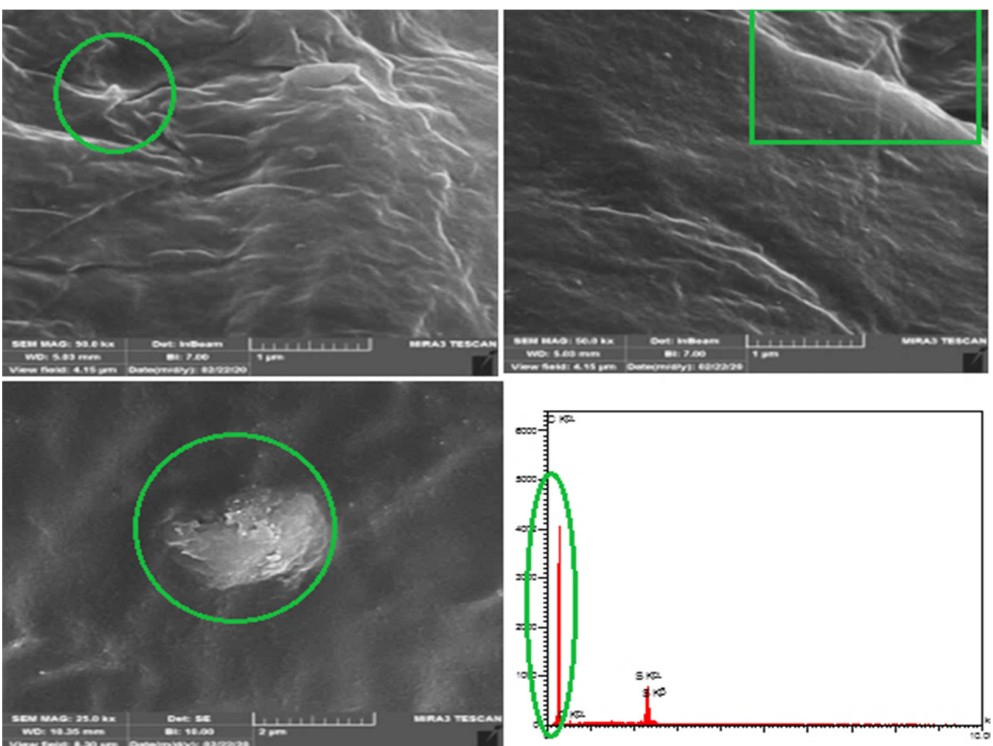

**Figure 12.** FE-SEM and EDX for asphalt (as received).

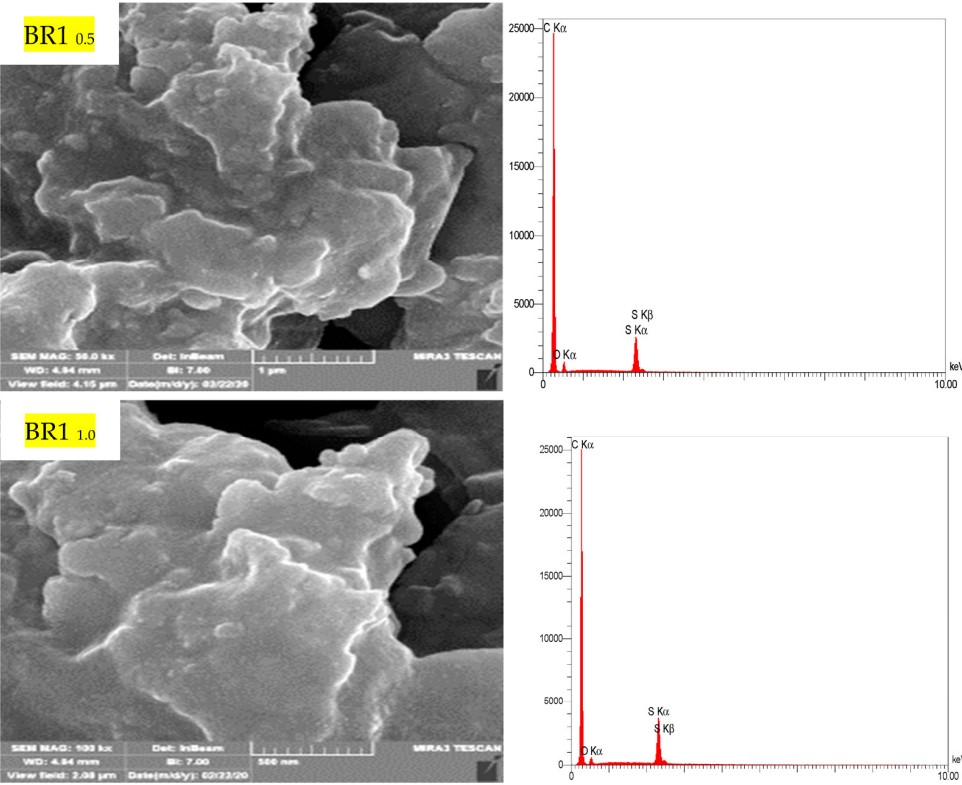

**Figure 13.** FE-SEM and EDX for BR-1 sample modified by 0.5% and 1.0% by weight of $AlCl_3$, using 1.0% of SRT before and after the aging procedure (at 10 min), treated at 360 °C.

## 4. Conclusions

The rheological properties of Qayara asphalt were improved by using spent rubber tires (SRT) with different percentages of anhydrous aluminum chloride. All required measurements were made on the modified asphalt one year after the modification process to understand the effect of aging conditions, which provided good rheological properties of the modified bitumen binder that is aging resistant. Eventually, the microwave energy was used to reduce the modification time from hours to minutes. Spent tire rubbers (STR) show a positive impact on the rheological properties of asphalt composites due to the filling effect and improve the microstructure of new proposed mixtures. In addition, microwave energy decreases the pollutant gas content that may evolve during the treatment and reduces environmental air pollution, leading to better life cycle assessment. It was clear that thermal sensitivity enhanced the penetration index (PI), which was a good value between −2.0 and +2.0. The increase in the thermal stability of output shows better workability during curing ages. The substation of a small amount of rubber 1.0% wt. percentage into asphalt composites with 0.5% catalyst (BR1) makes the system more homogenous and processing, fundamentally for asphalt resistance to weather condition improvement. More research is needed to reach sustainability in raw materials, as asphalt is a high-consumption material worldwide.

**Author Contributions:** Data curation, K.A.O. and R.Y.G.; Formal analysis, K.A.O.; Investigation, R.Y.G. and M.A.A.; Methodology, M.A.A.; Resources, K.A.O., R.Y.G. and M.A.A.; Writing—original draft, M.A.A. All authors have read and agreed to the published version of the manuscript.

**Funding:** This research received no external funding.

**Data Availability Statement:** Not applicable.

**Conflicts of Interest:** The authors declare no conflict of interest.

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
