# Peer review of "Study of the Effect of Modification of Asphalt on the Rheological Properties Employing Microwave Radiation—An Aging Study"

_recycling, doi:10.3390/recycling8050065_

Round 1

Reviewer 1 Report (Previous Reviewer 2)

The paper has been improved to a certain extent after the author's revision, but there are still some problems:

The results and discussion section of the article still needs to be revised, not just to list the data, but to analyze the data results and then draw regular conclusions.

Author Response

Study of the Effect of Modification of Asphalt on the Rheological Properties Employing Microwave Radiation - An Aging Study

   Reviewer #1

  • The paper has been improved to a certain extent after the author's revision: 
    • “Thank you very much for taking the time to put forward valuable opinions on our paper. They are of Great guidance significance to our paper writing and scientific research. The following explanations are made under your opinions, and highlighted in yellow inside the manuscript. “.
  1. The results and discussion section of the article still needs to be revised, not just to list the data, but to analyze the data results and then draw regular conclusions.

 Response:

  • Well noted ……. We act accordingly.
  • Title has been changed before submission, in order to be a complete sentence.
  • We revise the results and discussion section and the changes are highlighted in yellow

Thanks in advance

  Author

Reviewer 2 Report (New Reviewer)

1. The manuscript has a long abstract which contains many details, please make it more concise.

2. The exact findings of this study should be included in the abstract.

3. A comprehensive review was made in the Introduction on the current studies. One comment is that the authors cite more up-to-date references.

4. There are some small grammatical or writing errors. Please check and modify them. 

Author Response

  • “Thank you very much for taking the time to put forward valuable opinions on our paper. They are of Great guidance significance to our paper writing and scientific research. The following explanations are made under your opinions, and highlighted in yellow inside the manuscript. “.

 Comment: 1

  1. The manuscript has a long abstract which contains many details, please make it more concise.

 Response: 1

  • Dear Professor……. Well noted.
    • See abstract P1, L18-19.
    • Thereafter, the necessary measurements of the modified asphalt were taken to determine the extent to which it can be used for paving operations

 Comment: 2

  1. The exact findings of this study should be included in the abstract.

Response: 2

  • Dear Professor……. Well noted, we act accordingly.
    • See abstract P1, L20-24.
    • The measurements performed are plasticity, penetration, softening point, and penetration index. The previously mentioned measurements were also made on the modified asphalt one year after the modification process to understand the effect of aging conditions. The microstructure and thermodynamics have been characterized by FE-SEM and EDX measurements. This study provides good rheological properties of the modified bitumen binder that is aging resistant.

Comment: 3

  1. A comprehensive review was made in the Introduction on the current studies. One comment is that the authors cite more up-to-date references.

Response: 3

  • Dear Professor……. Well noted, we act accordingly
    • We cite more up-to-date references in the paper.

 Comment: 4

  1. There are some small grammatical or writing errors.

Response: 4

  • Dear Professor……. Well noted, we act accordingly
    • Rescan the manuscript and correct all grammatical or clerical errors.

Thanks in advance

Authors 

Reviewer 3 Report (New Reviewer)

The subject addressed in the manuscript is worthy of investigation, especially for the rheological properties and aging resistant of rubber tire modified asphalt. However, the manuscript has some minor problems, which need to be minor revised before it is submitted to the journal again:

(1) Please include some of the key results in the abstract.

(2) The quality of figures and references need to be further improved.

(3) You need to improve the conclusion section. May bullets would be better.

Author Response

The subject addressed in the manuscript is worthy of investigation, especially for the rheological propertiesand aging resistant of rubber tire modified asphalt. However, the manuscript has some minor problems, which need to be minor revised before it is submitted to the journal again:

  • “Thank you very much for taking the time to put forward valuable opinions on our paper. They are of Great guidance significance to our paper writing and scientific research. The following explanations are made under your opinions, and highlighted in yellow inside the manuscript. “.

 Comment: 1

  1. Please include some of the key results in the abstract.

  Response: 1

  • Dear Professor……. Well noted, we act accordingly.
    • See abstract P1, L20-24.
    • The measurements performed are plasticity, penetration, softening point, and penetration index. The previously mentioned measurements were also made on the modified asphalt one year after the modification process to understand the effect of aging conditions. The microstructure and thermodynamics have been characterized by FE-SEM and EDX measurements. This study provides good rheological properties of the modified bitumen binder that is aging resistant.

  • Comment: 2
  1. The quality of figures and references need to be further improved.
  • Response: 2
  • Dear Professor……. Well noted, we act accordingly
    • We re-structure/quality for all the figures

Comment: 3

  1. You need to improve the conclusion section. May bullets would be better.

Response: 3

  • Dear Professor……. Well noted, we act accordingly
    • See conclusion section P14, L325-332
    • It was clear that thermal sensitivity enhanced the penetration index (PI) that was a good value between -2.0 and +2.0. The increase in the thermal stability of output and shows better workability during curing ages. The substation of a small amount of rubber 1.0% wt. percentage into asphalt composites with 0.5% catalyst (BR1) makes the system more homogenous and processing. Fundamentally for asphalt resistance to weather condition improvement. More research needed to reach sustainability in raw materials, as asphalt is a high-consumption material worldwide

Thanks in advance

Authors 

This manuscript is a resubmission of an earlier submission. The following is a list of the peer review reports and author responses from that submission.

Round 1

Reviewer 1 Report

Title: Recycling of Spent Tire Rubber Enhanced Asphalt Rheological Properties Employing Microwave Radiation - An Aging Study

Number:2000320

In this paper, the rheological properties of Qayara asphalt were improved by using spent rubber tire (SRT), with different percentages of anhydrous aluminum chloride. Some measurements were made on the hub asphalt one year after the modification process to understand the effect of aging conditions, which provided good rheological properties of the hub bitumen binder that is aging resistant. In general, the paper is  innovative enough, but it needs to be revised to publish. Specific comments are as follows:

1. The title should be a complete sentence summarizing the main content of the research. Suggest recondensing the title.

2. Some of the professional vocabularies in Table 1 are not professional enough. It is suggested to be revised.

3. In the whole paper, the author only used a few indicators of the basic physical properties to analyze the rheological properties of the asphalt binder. It is not accurate and convincing. It is suggested to supplement other rheological indicators for detailed analysis.

4. For the performance test of the asphalt mixture, the author only carried out the Marshall test, which was not convincing. It is suggested to evaluate the comprehensive performance of the mixture by supplementary test.

Reviewer 2 Report

Point 1: Many scholars have studied the advantages of polymer modified asphalt and the microwave radiation of waste tire rubber powder, but there seems to be no innovation in this research.

Point 2: The abstract part is not clear and detailed, although the microstructure and thermodynamics of 20 kinds of materials are characterized by FE-SEM and EDX measurement, the test results are not explained, and the positive influence of the test results on the practical application of engineering is not summarized.

Point 3: The key words are not fine and sufficient, especially "He" appears in the third key word, which is not allowed in a standard scientific research article. In addition, the key words are only materials and experimental means, and there are no key words related to the experimental conclusion.

Point 4: The introduction only lists the existing research results of some scholars without detailed analysis, and does not discuss the necessity and innovation of this scientific study.

Point 5: The Materials and Methods section is relatively clear and detailed, but some writing errors were found. In lines 102,104 and 107 (not all listed), the figures or tables should be followed by figures without brackets; almost all figure and table titles are not centered; in line 182, "Figure7-", the The "-" is redundant, similar to lines 205.209 and 212, etc.

Point 6: In the conclusion and discussion section, the authors do not seem to analyze and discuss the results of the experiment, but simply present the icons, which does not reflect the innovation and uniqueness of this study. The same is true for the conclusion section, where the oversimplified sentence"The microwave energy was used to reduce the modification time from hours to minutes" is not a true conclusion of the study. The authors should add the conclusion.

Reviewer 3 Report

This manuscript studied the STR enhanced asphalt rheological properties by employing microwave radiation. Overall this is an interesting study. However, this manuscript shows the following:

  • Incomplete data. test data in figures show missing or poor controls
  • Inaccurate discussion/assumptions that are not supported by the FE-SEM observations.
  • Lack of proper structure/quality for figure presentations
  • Not enough of an advance or of enough impact for the journal

Also, please see attached pdf for other comments.
